# The Use of Machine Learning Algorithms and the Mass Spectrometry Lipidomic Profile of Serum for the Evaluation of Tacrolimus Exposure and Toxicity in Kidney Transplant Recipients

**DOI:** 10.3390/biomedicines10051157

**Published:** 2022-05-17

**Authors:** Dan Burghelea, Tudor Moisoiu, Cristina Ivan, Alina Elec, Adriana Munteanu, Ștefania D. Iancu, Anamaria Truta, Teodor Paul Kacso, Oana Antal, Carmen Socaciu, Florin Ioan Elec, Ina Maria Kacso

**Affiliations:** 1Clinical Institute of Urology and Renal Transplantation, 400006 Cluj-Napoca, Romania; dr.danburghelea@gmail.com (D.B.); tmoisoiu@gmail.com (T.M.); dralinaelec@gmail.com (A.E.); munteana2@yahoo.com (A.M.); antal.oanna@gmail.com (O.A.); 2Department of Urology, “Iuliu Hatieganu” University of Medicine and Pharmacy Cluj-Napoca, 400012 Cluj-Napoca, Romania; 3Biomed Data Analytics SRL, 400696 Cluj-Napoca, Romania; 4“Regina Maria” Hospital, 400117 Cluj-Napoca, Romania; dr.ivancristina@gmail.com; 5Faculty of Physics, Babeș-Bolyai University, 400084 Cluj-Napoca, Romania; stefania.iancu22@yahoo.ro; 6Research Center for Functional Genomics, Biomedicine and Translational Medicine, “Iuliu Hațieganu” University of Medicine and Pharmacy Cluj-Napoca, 400337 Cluj-Napoca, Romania; dr.amma.truta@gmail.com; 7Department of Nephrology, “Iuliu Hatieganu” University of Medicine and Pharmacy Cluj-Napoca, 400012 Cluj-Napoca, Romania; teokacso@gmail.com (T.P.K.); inakacso@yahoo.com (I.M.K.); 8Department of Anesthesiology, “Iuliu Hatieganu” University of Medicine and Pharmacy Cluj-Napoca, 400012 Cluj-Napoca, Romania; 9Faculty of Food Science and Technology, University of Agricultural Science and Veterinary Medicine Cluj-Napoca, Calea Mănăştur 3–5, 400372 Cluj-Napoca, Romania; csocaciudac@gmail.com

**Keywords:** tacrolimus, kidney transplant, metabolomic biomarkers, nephrotoxicity, machine learning, kidney graft function, liquid chromatography–mass spectrometry

## Abstract

Tacrolimus has a narrow therapeutic window; a whole-blood trough target concentration of between 5 and 8 ng/mL is considered a safe level for stable kidney transplant recipients. Tacrolimus serum levels must be closely monitored to obtain a balance between maximizing efficacy and minimizing dose-related toxic effects. Currently, there is no specific tacrolimus toxicity biomarker except a graft biopsy. Our study aimed to identify specific serum metabolites correlated with tacrolinemia levels using serum high-precision liquid chromatography–mass spectrometry and standard laboratory evaluation. Three machine learning algorithms were used (Naïve Bayes, logistic regression, and Random Forest) in 19 patients with high tacrolinemia (8 ng/mL) and 23 patients with low tacrolinemia (5 ng/mL). Using a selected panel of five lipid metabolites (phosphatidylserine, phosphatidylglycerol, phosphatidylethanolamine, arachidyl palmitoleate, and ceramide), Mg^2+^, and uric acid, all three machine learning algorithms yielded excellent classification accuracies between the two groups. The highest classification accuracy was obtained by Naïve Bayes, with an area under the curve of 0.799 and a classification accuracy of 0.756. Our results show that using our identified five lipid metabolites combined with Mg^2+^ and uric acid serum levels may provide a novel tool for diagnosing tacrolimus toxicity in kidney transplant recipients. Further validation with targeted MS and biopsy-proven TAC toxicity is needed.

## 1. Introduction

Kidney transplantation (KTx) is considered the gold-standard treatment for end-stage renal disease, providing a better quality of life and a higher survival rate than chronic dialysis [1,2].

The clinical management of patients who undergo renal transplants is challenging. Although surgical techniques have undergone significant advances in the past decade, the complexity of the immunological mechanisms involved, the poor quality of graft allocation strategies, and many knowledge gaps with respect to the personalization of immunosuppression therapies are responsible for significant differences in terms of graft survival [3,4]. To fill these gaps, tacrolimus (TAC), a calcineurin inhibitor (CNI), was introduced as a first-line chronic immunosuppression treatment alongside mycophenolate and steroid drugs. Even though TAC is 10–100 times more efficient than cyclosporine (a member of the CNI family) [5,6], the first two major clinical trials after the introduction of TAC in KTx show high rates of acute rejection (AR) in the first year after KTx (14–31%) [7,8], alongside a high percentage of toxicity events (39% nephrotoxicity, 6% neuropathy, 10% paresthesia, 13% diabetes mellitus, and 36% hyperglycemia) [7].

For each transplanted patient, it is well known that maintaining the perfect TAC blood concentration is a challenge due to pharmacodynamic and pharmacokinetic variations. Therefore, this translates into a narrow therapeutic window for TAC, which may put patients at risk for toxicity or graft rejection [9].

The current strategy for differentiating TAC nephrotoxicity and AR is based on TAC levels and serum creatinine evaluation. Unfortunately, these two parameters are suboptimal since patients with normal TAC levels may face increased creatine levels because of nephrotoxicity or AR. In this case, the current protocols call for the use of a graft biopsy, an invasive procedure, with TAC nephrotoxicity being an exclusion diagnostic [10]. For this reason, the search for novel biomarkers associated with TAC nephrotoxicity is intense, with most studies focusing on the identification of protein biomarkers in urine and serum, such as urinary NGAL, cystatin C, glutathione transferase, serum β-2 microglobulin, and α-1 microglobulin [11]. Unfortunately, none of them have managed to translate into practice [10]. 

Metabolomics, via high-performance liquid chromatography coupled with mass spectrometry (UHPLC–MS), offers new opportunities for the high-throughput measurement of large numbers of small molecules (<1500 Da), profiling the metabolites and identifying the most relevant ones by multivariate statistics, with possible clinical relevance.

Aside from the polar metabolites mentioned above, increasing consideration has recently been paid to the impact of tacrolimus on the lipid metabolism. For these reasons, in this study, we aimed to evaluate serum metabolomic and biochemical profiles of kidney graft recipients with outranged TAC levels (<5 ng/mL vs. >8 ng/mL), using untargeted lipidomic investigation by UHPLC–MS and machine learning algorithms; this strategy could be used for differential diagnostics of TAC toxicity (due to high TAC levels) and AR (due to insufficient TAC levels).

## 2. Materials and Methods

For this prospective transversal study, we enrolled 135 stable (defined as a creatinine level variation below 25% of the mean creatinine value), consecutive outpatients who underwent a KTx in our institution and for whom we performed standard follow-up between May 2020 and July 2020 at the Clinical Institute of Urology and Renal Transplantation Cluj-Napoca. The inclusion criteria were patients diagnosed with end-stage renal disease who underwent kidney transplantation with a TAC-based immunosuppressive therapy protocol (Advagraf 0.075–0.3 mg/kg/day), at least six months after the surgery. Patients with autoimmune diseases and those who developed lymphoproliferative disorders after kidney transplantation were excluded.

Clinical examinations, standard hematology, biochemistry panels, and tacrolinemia analyses were performed on all patients. Patients with outranged tacrolinemia (<5 or >8 ng/mL) were selected for metabolic profiling, as depicted in the workflow (Figure 1).

We defined the patients with outranged TAC levels over 8 ng/mL as the TAC high-level group (H-TAC) and the patients who had low-level TAC under 5 ng/mL as the TAC low-level group (L-TAC).

### 2.1. Sample Processing

Blood samples were collected from patients at 24 h post-dosing of Advagraf, just prior to the next dose. Patients had fasted for a minimum of 8 h.

The serum samples were collected by vein puncture into vacutainer tubes without anticoagulants. The blood serum was separated by centrifugation at 2000× *g* for 10 min, and aliquots of 1 mL were stored at −80 °C until analysis was performed. An 0.8 mL mixture of methanol and acetonitrile (1:1) was added to 0.2 mL of serum to precipitate the protein content; the mixture was then vortexed for 1 min, maintained at 4 °C for 6 h, and then vortexed again for 1 min. After mixing, the vials were centrifuged at 12,500× *g* for 5 min, and the supernatant was collected and filtered through 0.2 µm nylon filters.

### 2.2. Laboratory Tests

Tacrolinemia was measured using semi-automated electrochemiluminescence immunoassays using the ArchitectPlus CI4100 automatic analyzer. Prior to the initiation of the automated Architect sequence, a manual pretreatment step was performed in which the whole blood sample was extracted with a precipitation reagent and centrifuged. The supernatant was decanted into a Transplant Pretreatment Tube, which was placed onto the Architect iSystem [12,13].

The standard laboratory evaluation using the ArchitectPlus CI4100 automatic analyzer included serum cholesterol, triglycerides, glycemia, aspartate aminotransferase (ASAT), alanine aminotransferase (ALAT), gamma-glutamyl transferase (GGT), amylases, total proteins (TP), potassium (K^+^), sodium (Na^2+^), chloride (Cl^−^), ionized calcium (Ca^2+^), magnesium (Mg^2+^), and uric acid (UA). Renal function was determined via the estimated glomerular filtration rate (eGFR), using the creatinine-based CKD-EPI equation [14,15].

The metabolomic serum profile was analyzed using high-precision liquid chromatography (UHPLC)–mass spectrometry (MS) analysis.

The UHPLC–MS analysis was performed on a Bruker Daltonics MaXis Impact (Bruker GmbH, Bremen, Germany) device that comprised a Thermo Scientific UHPLC UltiMate 3000 system with a Dionex Ultimate quaternary pump delivery and ESI+-QTOF-MS detection device on a C18 reverse-phase column (Acuity, UPLC C18 BEH, Dionex) (5 µm, 2.1 × 75 mm) at 25 °C and with a flow rate of 0.3 mL/min. The injection volume was 5.0 µL. The mobile phase was represented by a gradient of eluent A (water containing 0.1% formic acid) and eluent B (methanol: acetonitrile 1:1, containing 0.1% formic acid). The gradient system consisted of 99% A (min 0), 70% A (min 1), 40% A (min 2), 20% A (min 6), and 100%B (min 9–10), followed by 5 min with 99% A. The total running time was 15 min. The MS parameters were set for a mass range between 50 and 1000 Da. The nebulizing gas pressure was set at 2.8 bar, the drying gas flow at 12 L/min, and the drying gas temperature at 300 °C. Before each chromatographic run, a calibration with sodium formate was performed. Instrument control and data processing were performed using the TofControl 3.2, Hystar 3.2, and Data Analysis 4.2 software packages provided by Bruker Daltonics.

### 2.3. Statistical Methods

The metabolites identified by UHPLC–MS and from blood tests were ranked based on their ability to discriminate between the high- and low-tacrolinemia groups using the *t*-test feature selection method. The Student’s *t*-test was applied for each metabolite, and the significance level was set to 0.05. The 5 statistically significant metabolites, along with the statistically significant blood parameters, were selected for further analysis. The classification accuracy for high and low tacrolinemia based on each significant metabolite was evaluated using a receiver operating characteristic curve for which the area under the curve (AUC) was calculated.

To quantitatively evaluate the multivariate classification power yielded by the biochemical blood parameters and the 5 significant metabolites, 3 independent machine learning algorithms (naive Bayes, logistic regression, and Random Forest) were trained to discriminate between the H-TAC and L-TAC groups. For logistic regression, the regularization type was set to Lasso and the C parameter to 140. For the Random Forest analysis, 10 trees were implemented. All the models were cross-validated using the leave-one-out method.

The inputs for the machine learning algorithms were either the biochemical blood parameters alone, the selected five metabolites alone, or all combined. For the classification based on the combined biochemical blood parameters and the five metabolites, data were normalized to unity prior to classification. The performance of the classification was assessed in terms of the AUC derived using receiver operating characteristic (ROC) analysis, classification accuracy, F1 score, precision, and recall. The quality performance metrics were represented as the average of the values from each repetition of the cross-validation.

Next, a principal component analysis (PCA) was performed to explore the dataset; the 5 metabolites along with the biochemical blood levels were used as inputs. For a better representation of the capacity to differentiate the TAC level of the experimental model, PCA was used to reduce the data dimensionality. The relationship between the number of PCs and the explained variability in the original dataset is presented in Appendix A.

For the correlation analysis, we used the Pearson correlation coefficient.

All statistical analyses were performed using the Quasar-Orange software (Bioinformatics Laboratory of the University of Ljubljana) [16,17].

The study was approved by the Ethics Committee of the Clinical Institute of Urology and Kidney Transplantation, Cluj-Napoca, No. 2/2020, and by the Ethics Committee of the Iuliu Hatieganu University of Medicine and Pharmacy in Cluj-Napoca, No. 285/2020. Written informed consent was obtained from all patients following the rules and principles of the Helsinki Declaration.

## 3. Results

Our dataset comprised 42 patients with outranged tacrolinemia levels; 19 and 23 patients had high (over 8 ng/mL) and low (under 5 ng/mL) levels of tacrolinemia, respectively, after KTx. The demographic data of the two groups are presented in Appendix A.

From the biochemistry panel, only Mg^2+^ and UA levels were changed to a statistically significant extent between the H-TAC and L-TAC groups, as depicted in Table 1.

Using UHPLC–MS, 336 metabolites were identified (Appendix A), from which only five were significantly related to over-ranged blood tacrolimus levels: phosphatidylserine 44:8 (PS), phosphatidylglycerol 36:6 (PG), phosphatidylethanolamine 36:4 (PE), arachidyl palmitoleate C36:1 (AP), and ceramide t18:0/22:0(2OH) (CER).

The Student’s *t*-test and ROC analysis were used for the selection of the five statistically relevant metabolites (Table 2).

The violin plots of the UHPLC–MS counts for the five selected metabolites and the serum concentrations of Mg^2+^ and UA are presented in Figure 2.

Next, three machine learning algorithms (naïve Bayes, logistic regression, and Random Forest) were applied to evaluate the discrimination of the H-TAC and L-TAC groups using Mg^2+^, UA, and the five metabolites. (Table 3, Table 4 and Table 5, Figure 3c).

The distribution of score values following PCA of the H-TAC and L-TAC groups’ metabolic profiles (PC1 and PC2) are represented in Figure 3a and show the clustering tendency of the H-TAC and L-TAC groups. Figure 3b presents the loading plot for PC1 and PC2, showing that CER, PE, PG, and PS are the variables that contribute the most to PC1, and UA and Mg^2+^ to PC2. Moreover, a negative correlation between Mg^2+^ and AP on the one hand and CER, PE, PG, and PS on the other hand is determined by the loading plot of PC1. By examining the score plot and loading plots pf PC1 and PC2, it is observed that the H-TAC group with positive values for PC1 has high values of CER, PE, PG, and PS, while the L-TAC group shows high values for Mg^2+^, and AP.

Next, to evaluate the relation between the selected metabolites, we performed a correlation analysis between each metabolite, Mg^2+^, and UA, using the Pearson correlation coefficient (Figure 4). We identified a statistically significant (*p* < 0.05) positive and moderate correlation between AP and Mg^2+^ and between CER and PE, and a high correlation between PE and PS. Between AP and PS, PE, and AP, we found statistically significant, moderate, and negative correlations, respectively.

## 4. Discussion

In this current study, we use untargeted UHPLC–MS serum profiling and routine biological evaluation of the serum for the diagnosis of TAC toxicity. From 135 stable consecutive KTx recipient patients for whom TAC serum level was evaluated, we selected 19 patients with low tacrolinemia (<5 ng/mL) (L-TAC) and 23 patients with high tacrolinemia (>8 ng/mL) (H-TAC) (Figure 1).

The currently recommended standard immunosuppression therapy for patients undergoing KTx comprises CNI (preferably TAC because of its higher efficacy) combined with mycophenolate and steroids [6,18]. Because of the narrow therapeutic window of TAC therapeutic protocols, the transplanted patients are potentially at risk of underexposure and allograft rejection, or on the contrary, overexposure, and toxicity [19]. Consequently, the determination of serum TAC and creatinine is insufficient to assess the optimal systemic exposure. 

From a pharmacokinetic point of view, tacrolimus is metabolized in the liver but also in the gut and kidney; this process is mediated by the phase I oxidase system via CYP3A4/5 [20,21] and the phase II metabolism by demethylation, glucuronidation, sulfation, acetylation, and conjugation. The metabolites are present in low concentrations in the blood and have minor pharmacological activity when compared to tacrolimus itself and are of minor clinical relevance [22].

Regarding the liver metabolization of TAC, its active metabolite, 6-mercaptopurine (6-MP), is metabolized via three different metabolic pathways. It can be inactivated by thiopurine methyltransferase to 6-metylmercaptopurine or by xanthine oxidase to 6-thiouric, or it can be activated to 6-thioguanine nucleotide, which explains its therapeutic availability. TAC is primarily metabolized by the CYP3A enzyme system, which includes CYP3A5, CYP3A4, CYP3A7, and CYP3A43, and it is expressed in the small intestine, liver, and kidney [9,23,24]. 

In order to optimize the TAC therapeutic response, i.e., to minimize subtherapeutic and supratherapeutic TAC exposure in the immediate post-transplant phase, recent research studies aim to identify novel biomarkers correlated with TAC exposure; such biomarkers might provide an accurate algorithm to predict an individual’s TAC starting dose and the therapeutic dosage required to improve clinical outcome in kidney transplant patients [25,26,27].

Experimental animal models show that there is a change in the metabolomic profile of urine after the administration of cyclosporine (decreased levels of succinate, citrate, and alpha-ketoglutarate, and increased levels of taurine) [28]. In contrast, time-related studies showed that, at 28 days after the administration of cyclosporine, there is a reduction in Krebs cycle intermediates and trimethylamine-N-oxide concentrations, whereas acetate, lactate, trimethylamine, and glucose concentrations increase [29]. These results were recently validated in humans [30]. Regarding TAC, to our knowledge, there is only one published study by Klepacki et al., who validated a panel of ten urine metabolites used in cyclosporine studies (glucose, hippurate, lactate, oxoglutarate, sorbitol, succinate, TMAO, UA, citrate, and creatinine) using targeted MS [31]. Interestingly, the result showed that, after three months, the level of the selected metabolites returned to normal, except for oxoglutarate, lactate, and uric acid [31]. It is unknown whether the level of the selected metabolite would change in case of TAC toxicity.

In our study, all the metabolites that differed between the H-TAC and L-TAC groups were components of the lipid metabolism. Kim et al. showed that the level of lipids increases after transplantation, especially in patients treated with cyclosporine A. Using proton nuclear magnetic resonance, they identified LDL, VLDL CH3, lipid CH2CH2CO, lipid CH2C=C, lipid CH2CO, and lipid CH as being upregulated [32]. Immunosuppressive therapy in KTx patients leads to the accumulation of triglyceride-enriched VLDL and LDL, increasing the atherosclerotic and cardiovascular risk for these patients [33,34,35].

Mechanisms of hyperlipidemia associated with CNI were more extensively studied with cyclosporine, which interferes with the binding of LDL cholesterol to the LDL receptor, bile acid synthesis, and 26 hydroxylase enzymes. In addition, cyclosporine is highly lipophilic and transported within the core of LDL cholesterol particles. In the process, it may change the molecular configuration of LDL. TAC generally provides a safer lipidic profile than cyclosporin, as cyclosporin new-onset hyperlipidemia remissions were reported after switching to TAC-based immunosuppression. However, the in-depth interactions of TAC with the lipid metabolome have not been extensively studied or applied in clinical practice [34,36,37]. 

Phosphatidyl glycerol, phosphatidylserine, phosphatidylethanolamine, and phosphatidylinositol (PI)-precursor of phosphatidylinositol 3-phosphate (PI3P) are cell membrane glycerophospholipids derived from the same glycerol backbone, namely diacylglycerol (DAG), as presented in Figure 5, and have different roles in biological membranes [38,39]. The pathway between CNI and induced changes in glycerophospholipid synthesis is still being studied, but CNI seems to inhibit the expression of phosphoinositide-3-kinase (PI3K) and other protein kinases as a mechanism of inducing nephrotoxicity [40]. In a recently published article, Karolin et al. showed that the CNI-induced nephrotoxic effect is obtained by an independent pathway from the known nuclear-activated T-cell (N-FAT) mechanism [40]. Thus, CNI seems to inhibit the expression of many protein kinases, including PI3K. The authors showed that blocking protein kinases in the tubular epithelial cells of the nephron leads to the increased expression of fibroblast growth factor–inducible 14 (Fn14), the receptor of TWEAK (TNF-related weak inducer of apoptosis), a key molecule involved in fibrosis and apoptosis in the kidney and renal graft [41]. Because CNI inhibits PI3K expression, there seems to be an accumulation of its substrate, PI3P and DAG, which are, in turn, metabolized in Phosphatidyl glycerol, phosphatidylserine, and phosphatidylethanolamine (Figure 5), in line with our result that shows increased levels of PG, PS, and PE in the H-TAC group compared to the L-TAC group (Table 2). Further investigation of these hypotheses is needed.

Palmitic acid derivates were identified by our study as occurring in lower amounts in the H-TAC group, consistent with upregulation of the glycerophospholipid pathway to the detriment of the fatty acid/triglyceride pathway [42]. 

Ceramides are central molecules of the sphingolipid metabolism, with essential bioactive implications in cell processes such as apoptosis, necrosis, and autophagy-dependent cell death [43]. Increased levels of ceramides are strongly connected with the deterioration of pancreatic beta-cell function, insulin sensitivity, vascular reactivity, and mitochondrial metabolism; therefore, there are studies showing their presence in heart disease, atherosclerosis, hepatic disease, insulin resistance, and diabetes [44]. Concerning renal disease, ceramides and their metabolites are recognized as being part of the pathological mechanism in acute kidney injury, kidney cancer, polycystic kidney disease, and diabetic nephropathy [45]. In a recent study conducted on 760 patients, it was shown that increased levels of ceramides were on the direct axis of focal segmentary glomerulosclerosis [46]. These findings are in line with our results describing higher levels of ceramide t18:0/22:0(2OH) in the H-TAC group compared to the L-TAC group and also support a possible pathway for drug-induced nephrotoxicity. 

After transplantation, Mg^2+^ serum levels decrease in part because the immunosuppressive therapy, especially CNI, increases Mg^2+^ urinary excretion. One study found that hypomagnesemia was observed in 6.6% of patients undergoing TAC therapy. On the other hand, hyperuricemia is a common complication in organ transplant recipients and is frequently associated with chronic immunosuppressive therapy (including TAC treatment), even though the role of UA levels in the survival of kidney grafts remains controversial [47,48,49,50,51].

There are few studies exploring metabolomics in kidney transplant patients. Previous studies that investigated allograft rejection and CNI-related side effects found that the metabolites were mainly represented by sugars, inositol, and hippuric acid [52]. These studies differed from ours in terms of the design and treatment explored (TAC). 

Furthermore, metabolites such as tryptophan and arginine were previously identified as potential biomarkers for acute kidney injury with a high AUC when compared to creatinine; however, that study failed to associate direct toxicity with TAC [53]. 

There are several limitations to our study. The most important is the use of only untargeted MS because, for the mass implementation of the panel, cut-off values are mandatory. Additionally, there is no validation group with graft-biopsy-proven TAC toxicity. Another limitation is the lack of analysis regarding the time from transplantation since previously published studies have shown that concentration changes in the metabolites related to oxidative stress are time-related [31].

For future research, we will use targeted MS to help establish a cut-off value for these metabolites, followed by their validation on a larger group of KTx patients with biopsy-proven TAC toxicity. Hopefully, this will help improve clinical outcomes, graft survival rate, dose adjustment and quality of life but also will represent essential tools in guiding therapeutic strategies.

## 5. Conclusions

Using UHPLC–MS serum profiling and machine learning algorithms we proved that KTx patients with abnormal TAC levels exhibit a particular metabolomic signature that might help diagnose TAC toxicity without graft biopsy based on a panel of five lipid metabolites, serum Mg^2+^, and UA. Our results need to be further validated with targeted MS on larger cohorts with biopsy-proven TAC toxicity. 

## Figures and Tables

**Figure 1 biomedicines-10-01157-f001:**
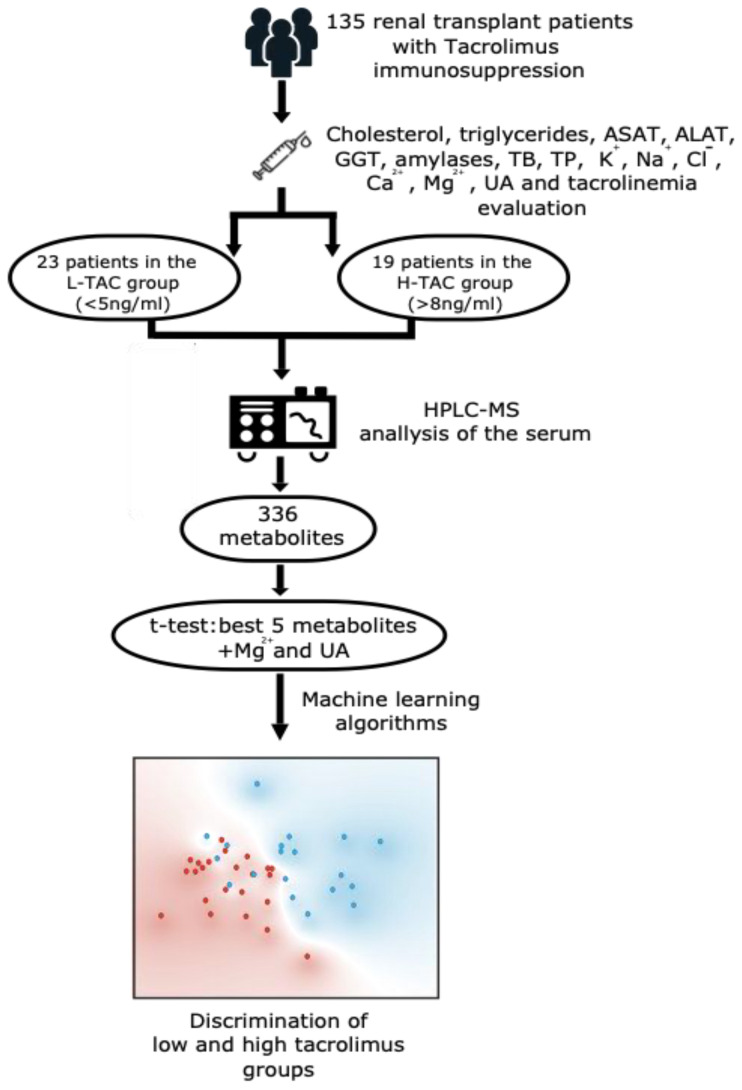
Workflow of the study. Abbreviations: ASAT—aspartate aminotransferase; ALAT—alanine aminotransferase; GGT—gamma-glutamyl transferase; TB—total bilirubin; TP—total proteins; K^+^—potassium; Na^+^—sodium; Cl^−^—chloride; Ca^2+^—ionized calcium; Mg^2+^—magnesium; UA—uric acid; L-TAC—low tacrolinemia group; H-TAC—high tacrolinemia group; UHPLC–MS—high-precision liquid chromatography–mass spectrometry analysis.

**Figure 2 biomedicines-10-01157-f002:**
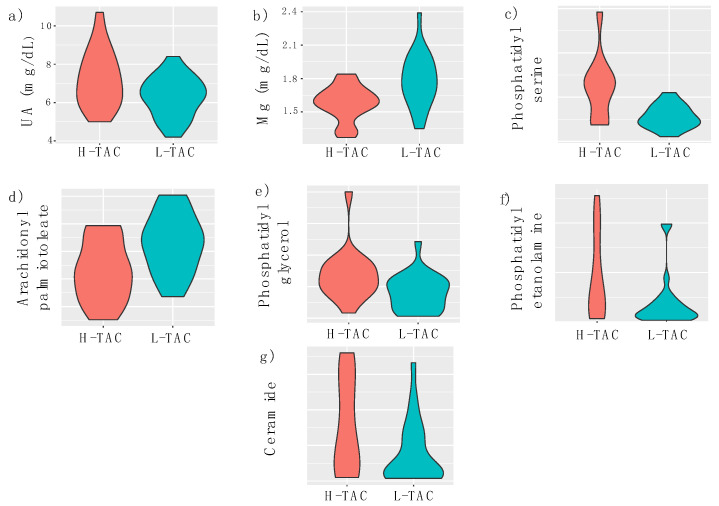
Violin plots of UA-uric acid (**a**), Mg^2+^-magnesium (**b**), phosphatidyl serine (44:8) (**c**), arachidyl palmitoleate (C36:1) (**d**), phosphatidyl glycerol (36:6) (**e**), phosphatidyl ethanolamine (36:4) (**f**), and ceramide (t18:0/22:0(2OH)) (**g**), for the high-tacrolinemia (H-TAC) and low-tacrolinemia (L-TAC) groups.

**Figure 3 biomedicines-10-01157-f003:**
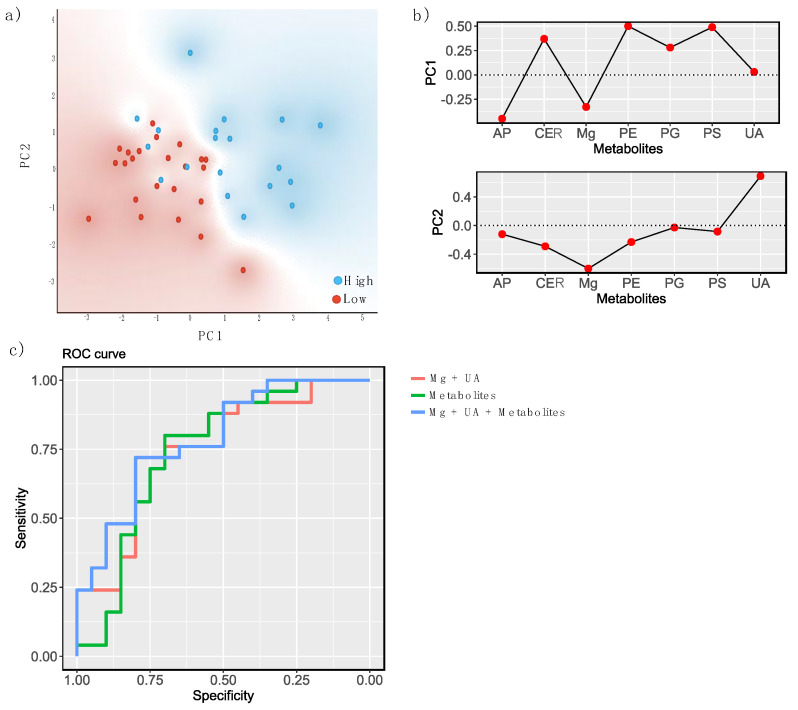
(**a**) The distribution of principal component (PC) score values (PC1 and PC2) of patients with metabolic profiles associated with low and high tacrolinemia. (**b**) Loading plots of the first two PCs yielded by PC analysis. (**c**) Head-to-head comparison of the receiver operating characteristic curves (ROC) for the classification accuracy yielded by magnesium, uric acid, the five metabolites (phosphatidylserine 44:8, phosphatidylglycerol 36:6, phosphatidylethanolamine 36:4, arachidyl palmitoleate C36:1, and ceramide t18:0/22:0(2OH)), and their combination using naïve Bayes analysis for supervised classification.

**Figure 4 biomedicines-10-01157-f004:**
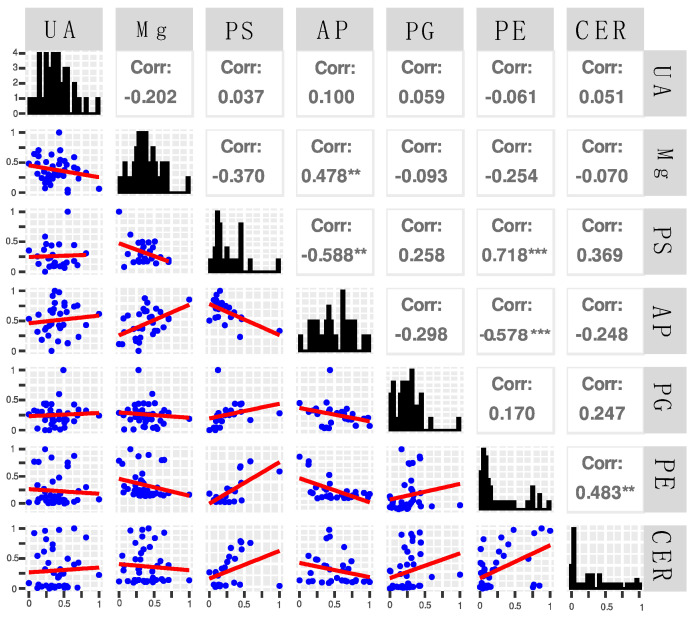
Correlation matrix and histogram between the metabolites, UA, and Mg^2+^. Abbreviations and legend: UA–uric acid; Mg^2+^–magnesium; phosphatidylserine 44:8 (PS), phosphatidylglycerol 36:6 (PG), phosphatidylethanolamine 36:4 (PE), arachidyl palmitoleate C36:1 (AP), and ceramide t18:0/22:0(2OH) (CER); Corr–Pearson correlation coefficient; ** *p* < 0.01; *** *p* < 0.001.

**Figure 5 biomedicines-10-01157-f005:**
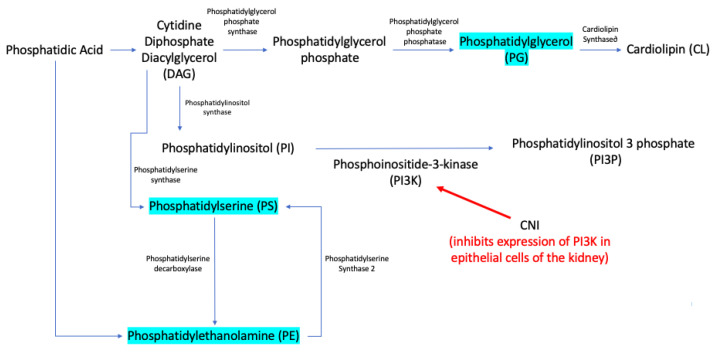
Lipid metabolic map.

**Table 1 biomedicines-10-01157-t001:** Student’s *t*-test and area under the curve for standard follow-up biochemical blood tests were used to discriminate between patients with low and high tacrolinemia.

Blood Tests	H-TAC Mean ± SD	L-TAC Mean ± SD	*t*-Test (p)	AUC
Cholesterol (mg/dL)	228 ± 82	208 ± 39	0.237	0.56
Triglycerides (mg/dL)	172 ± 97	147 ± 70	0.349	0.56
Potassium (mmol/L)	4.4 ± 0.4	4.4 ± 0.6	0.725	0.51
Amylases (U/L)	93 ± 24	85 ± 30	0.312	0.58
Creatinine (mg/dL)	1.6 ± 0.5	1.6 ± 0.9	0.318	0.59
ASAT (U/L)	20 ± 7.2	19 ± 8.2	0.470	0.56
ALAT (U/L)	27 ± 15	20 ± 15	0.051	0.67
GGT (U/L)	34 ± 22	30 ± 20	0.294	0.59
TB (mg/dL)	0.72 ± 0.37	0.73 ± 0.29	0.740	0.53
Glycemia (mg/dL)	101 ± 16	117 ± 73	0.638	0.54
Total proteins (mg/dL)	7 ± 0.4	6.8 ± 0.4	0.341	0.58
Ca^2+^ (mmol/L)	4.6 ± 0.52	4.4 ± 0.4	0.116	0.64
Cl^−^ (mmol/L)	107 ± 2.7	106 ± 3.9	0.814	0.52
Na^+^ (mmol/L)	142 ± 2.5	141 ± 1.9	0.111	0.66
Mg^2+^ (mmol/L)	158 ± 15.52	178.4 ± 23.65	0.001	0.7243
UA (mg/dL)	72.42 ± 14.95	63.09 ± 10.57	0.025	0.6636

**Table 2 biomedicines-10-01157-t002:** Student’s *t*-test and area under the curve for the significantly different metabolites used to discriminate between patients with low and high tacrolinemia. The mean levels of the metabolites represent peak UHPLC–MS intensities.

Metabolite	High Group Mean ± SD	Low Group Mean ± SD	*p*-Value	AUC
**PS (counts)**	245,714 ± 145,458	111,783 ± 52,986	0.01	0.818
**AP (counts)**	32,839 ± 11,132	42,818 ± 10,796	0.01	0.730
**PG (counts)**	273,380 ± 165,513	162,278 ± 115,156	0.02	0.724
**PE (counts)**	445,195 ± 419,624	197,051 ± 268,564	0.03	0.711
**CER (counts)**	464,002 ± 395,761	233,792 ± 263,222	0.03	0.807

Abbreviations: phosphatidylserine 44:8 (PS), phosphatidylglycerol 36:6 (PG), phosphatidylethanolamine 36:4 (PE), arachidyl palmitoleate C36:1 (AP), and ceramide t18:0/22:0(2OH) (CER).

**Table 3 biomedicines-10-01157-t003:** Head-to-head comparison of the area under the curve results for the classification accuracy yielded by magnesium and uric acid using three supervised classification algorithms.

Statistic Model	AUC	CA	F1	Precision	Recall
**Naïve Bayes**	0.621	0.578	0.579	0.585	0.577
**Logistic regression**	0.752	0.711	0.712	0.713	0.711
**Random Forest**	0.620	0.644	0.644	0.644	0.644

Abbreviations: AUC—area under the curve; CA—classification accuracy; F1 score; Precision-positive predictive value; Recall-sensitivity.

**Table 4 biomedicines-10-01157-t004:** Head-to-head comparison of the area under the curve results for the classification accuracy yielded by the five metabolites using three supervised classification algorithms.

Statistic Model	AUC	CA	F1	Precision	Recall
**Naïve Bayes**	0.750	0.667	0.667	0.683	0.667
**Logistic regression**	0.744	0.756	0.755	0.755	0.756
**Random Forest**	0.636	0.556	0.552	0.551	0.551

The five metabolites are phosphatidylserine 44:8, phosphatidylglycerol 36:6, phosphatidylethanolamine 36:4, arachidyl palmitoleate C36:1, and ceramide t18:0/22:0(2OH). Abbreviations: AUC—area under the curve; CA—classification accuracy; F1 score; Precision-positive predictive value; Recall-sensitivity.

**Table 5 biomedicines-10-01157-t005:** Head-to-head comparison of the area under the curve results for the classification accuracy yielded by magnesium, uric acid, and the five metabolites using three supervised classification algorithms.

Statistic Model	AUC	CA	F1	Precision	Recall
**Naïve Bayes**	0.799	0.756	0.756	0.764	0.756
**Logistic regression**	0.788	0.733	0.734	0.738	0.733
**Random Forest**	0.683	0.600	0.597	0.597	0.600

The five metabolites are phosphatidylserine 44:8, phosphatidylglycerol 36:6, phosphatidylethanolamine 36:4, arachidyl palmitoleate C36:1, and ceramide t18:0/22:0(2OH). Abbreviations: AUC—area under the curve; CA—classification accuracy; F1 score; Precision-positive predictive value; Recall-sensitivity.

## Data Availability

Not applicable.

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
