# Peer review of "The Use of Machine Learning Algorithms and the Mass Spectrometry Lipidomic Profile of Serum for the Evaluation of Tacrolimus Exposure and Toxicity in Kidney Transplant Recipients"

_biomedicines, 2022, doi:10.3390/biomedicines10051157_

Round 1

Reviewer 1 Report

This is an excellent work, outstanding by its scientific quality and meaningful results.

Suggestions:

1.Please emphasize in the conclusion section on the importance in the clinical practice of your results, even if this is an untargeted metabolomic analysis.

2. If appropriate, please do not introduce results data in the conclusion section.

Author Response

We thank the reviewer for their insightful suggestions. Please find below our point-by-point responses. The changes in the manuscript are highlighted by the track-change option of MS Word. We feel that our manuscript has now improved significantly with these comments, and we hope you find it suitable for publication in Biomedicines journal.

General comment

This is an excellent work, outstanding by its scientific quality and meaningful results.

Suggestions:

  1. Please emphasize in the conclusion section on the importance in the clinical practice of your results, even if this is an untargeted metabolomic analysis.

558-593: We thank the Reviewer for this recommendation. Regarding your suggestion, we added a paragraph mentioning the future perspective and the potential clinical implication: “Metabolomic profile, new cell and molecular biology and genomic acquisitions (molecular biomarkers) may become indicators of clinical outcome, survival rate, therapeutic response and life quality but also essential tools in guiding therapeutic strategies and improve clinical outcomes in KTx patients.”

  1. If appropriate, please do not introduce results data in the conclusion section.

Regarding the repetition of the results, in our opinion, it is important to mention them in a short way as a conclusion of our work.

Reviewer 2 Report

General comment

The manuscript entitled “The use of Machine Learning Algorithms and Mass Spectrometry Lipidomic Profile of Serum for the Evaluation of Tacrolimus Exposure and Toxicity in Kidney Transplant Recipients” aims to identify and evaluate specific serum metabolites correlated with Tacrolimus toxicity, utilizing three machine learning algorithms. The topic is interesting and, considering the paucity of studies reported, this work could further enrich the literature in merit. In addition to an overall English grammar and typos check, the manuscript requires a few major corrections regarding the fluency and the clarity of the work.

  • Major issues

INTRODUCTION

70-78: Those considerations would be more suitable in the discussion

86-88: I presume that for low levels of tacrolimus, the problem would be not the toxicity but the graft failure/rejection. Please be more clear in reporting the aim of your study and the potential implications.

MATERIALS AND METHODS

90: Please explain what do you mean by “stable”

153: How did you choose the best 5 statistically significant metabolites? There were other statistically significant metabolites not considered? and why?

RESULTS

240-250: Some of the results reported should be better placed in the methods, considering that explain the workflow reported in the figure 1.

DISCUSSION

268-284: This is a mere repetition of the introduction and results.

296-299: This is the main issue of the manuscript and the discussion should focus on this topic, reporting other similar studies in the literature. Considering that this would be the reason for not using Tacrolimus, your study could add further methods to improve the manageability of this drug. This should be also reported briefly in the introduction.

367: to this regard please also see: DOI: 10.1007/s15010-021-01706-6

CONCLUSION

Avoid the repetition of the results. Report future perspectives and potential clinical implications.

  • Minor issues

ABSTRACT

The abstract is not particularly clear regarding the methodology used and should be revised. In particular, it is not clear if graft biopsy was used. Considering that in the main text is clear the workflow, try to briefly summarize it also in the abstract.

RESULTS

194: This is not reported in the workflow, which is, fairly, generic.

228-231: Check sentences. Some were referred to a table probably.

REFERENCES

Try to use references not older than 2010.

Author Response

We thank the reviewers for their insightful suggestions. Please find below our point-by-point responses. The changes in the manuscript are highlighted by the track-change option of MS Word. We feel that our manuscript has now improved significantly with these comments, and we hope you find it suitable for publication in Biomedicines journal.

General comment

The manuscript entitled “The use of Machine Learning Algorithms and Mass Spectrometry Lipidomic Profile of Serum for the Evaluation of Tacrolimus Exposure and Toxicity in Kidney Transplant Recipients” aims to identify and evaluate specific serum metabolites correlated with Tacrolimus toxicity, utilizing three machine learning algorithms. The topic is interesting and, considering the paucity of studies reported, this work could further enrich the literature in merit. In addition to an overall English grammar and typos check, the manuscript requires a few major corrections regarding the fluency and the clarity of the work.

  • Major issues

INTRODUCTION

70-78: Those considerations would be more suitable in the discussion

448-460: We thank the Reviewer for this observation. According to the suggestion offered we moved the recommended paragraph to the discussion section:

“Experimental animal models show that there is a change in the metabolomic profile of urine after the administration of Cyclosporine (decreased levels of succinate, citrate, al-pha-ketoglutarate, and increased levels of taurine). In contrast, time-related studies showed that at 28 days after the administration of Cyclosporine, there is a reduction of Krebs cycle intermediates and trimethylamine-N-oxide concentrations, whereas acetate, lactate, trimethylamine, and glucose concentrations increase. These results were recently validated in humans. Regarding TAC, to our knowledge, there is only one published study by Klepacki et al. who validated a panel of ten urine metabolites used in Cyclosporine studies (glucose, hippurate, lactate, oxoglutarate, sorbitol, succinate, TMAO, UA, citrate, and creatinine) using targeted MS. Interestingly, the result showed that after three months, the level of the selected metabolites returned to normal, except for oxoglutarate, lactate and uric acid. It is unknown whether the level of the selected metabolite would change in case of TAC toxicity”

86-88: I presume that for low levels of tacrolimus, the problem would be not the toxicity but the graft failure/rejection. Please be more clear in reporting the aim of your study and the potential implications.

100-106: We thank the Reviewer for this comment. According to the recommendation, we modified the enouncement by making it more clear about the risks of overdosing and underdosing of Tacrolimus:

“Besides the polar metabolites mentioned above, recently, increasing considerations were related to the impact of tacrolimus on lipid metabolism. For these reasons, in this study we aimed the evaluation of serum metabolomic and biochemical profiles of kidney graft recipients with outranged TAC levels (<5 ng/mL vs >8 ng/mL), using untargeted lip-idomic investigation by UHPLC-MS and machine learning algorithms, that could be used for a differential diagnostics of TAC toxicity (due to high TAC levels) and AR (due to inefficient TAC levels).”

MATERIALS AND METHODS

90: Please explain what do you mean by “stable”

108-109: We thank the Reviewer for this suggestion. We explained what stable means: a creatinine level variation below 25% of the mean creatinine value.

153: How did you choose the best 5 statistically significant metabolites? There were other statistically significant metabolites not considered? and why?

203: We thank the Reviewer for this question. We are sorry for the miss-spelling; we find only 5 statistically significant metabolites, so we deleted the word “best”. They were chosen by the t-test value p<0.05.

RESULTS

240-250: Some of the results reported should be better placed in the methods, considering that explain the workflow reported in the figure 1.

227-231: We thank the Reviewer for this recommendation. As the reviewer mentioned above, we moved the paragraph “For a better representation of the capacity to differentiate the TAC level of the experimental model, PCA was used to reduce the data dimensionality. The relationship between the number of PCs and the explained variability in the original dataset is presented in Supplementary Figure 1.” in the methods chapter.

DISCUSSION

268-284: This is a mere repetition of the introduction and results.

340-344: We thank the Reviewer for this observation. Regarding this problem, we shorten the repetition of the introduction and the results. The only remaining phrase is:

“In this current study, we use untargeted UHPLC-MS serum profiling and routine bio-logical evaluation of the serum for the diagnosis of the TAC toxicity. From 135 stable consecutive KTx recipients’ patience for whom TAC serum level was evaluated, we se-lected 19 patients with low tacrolinemia (<5ng/ml) (L-TAC) and 23 patients with high tacrolinemia (>8 ng/mL) (H-TAC) (Figure 1).”

296-299: This is the main issue of the manuscript and the discussion should focus on this

topic, reporting other similar studies in the literature. Considering that this would be the reason for not using Tacrolimus, your study could add further methods to improve the manageability of this drug. This should be also reported briefly in the introduction.

We thank the Reviewer for this suggestion.

345-351: Listening to your advice we moved this statement upfront in the discussion chapter.

435-447: We added some paragraphs about some similar studies of the literature:

“In order to optimize tacrolimus TAC therapeutic response, minimize subtherapeutic and supratherapeutic tacrolimus exposure in the immediate post-transplant phase, recent research studies aim to identify novel biomarkers correlated with tacrolimus exposure, biomarkers which might provide an accurate algorithm to predict an individual’s tacrolimus starting dose and therapeutic dosage required to improve clinical outcome in kidney transplant patients.

Previous studies identified an algorithm for TAC  dosage based on cytochrome P450 (CYP)3A4 and CYP3A5, where genotype, body surface area, and age were covariates, but a multi combined biomarker strategy based on molecular biomarkers, clinical and biological biomarkers that include associated comorbidities, multi medication, age and ethnicity might provide an accurate algorithm which will optimize Tacrolimus dosage, therapeutic response, minimize side effects and improve clinical outcome and survival rates in kidney transplant recipients.”

83-86: As the reviewer recommended, we also introduced a short paragraph about this issue in the introduction:

“For each transplanted patient, it is well known that maintaining the perfect TAC blood concentration is a challenge due to the pharmacodynamics and pharmacokinetics variations. Therefore, this translates into a narrow therapeutic window of TAC, which may put at risk the patients for toxicity or graft rejection (9).”

367: to this regard please also see: DOI: 10.1007/s15010-021-01706-6

568: We thank the Reviewer for this suggestion. Reading the recommended article “Renal involvement in COVID-19: focus on kidney transplant sector” we didn’t find any significant correlation between this paper and the use of metabolomics in Tacrolimus induced toxicity phrase, maybe it’s a misunderstanding.

CONCLUSION

Avoid the repetition of the results. Report future perspectives and potential clinical implications.

558-593: We thank the Reviewer for this recommendation. Regarding your suggestion we added a paragraph mentioning the future perspective and the potential clinical implication:

“Metabolomic profile, new cell and molecular biology and genomic acquisitions (molecular biomarkers) may become indicators of clinical outcome, survival rate, therapeutic response and life quality but also essential tools in guiding therapeutic strategies and improve clinical outcomes in KTx patients.”

Regarding the repetition of the results, in our opinion, it is important to mention them in a short way as a conclusion of our work.

  • Minor issues

ABSTRACT

The abstract is not particularly clear regarding the methodology used and should be revised. In particular, it is not clear if graft biopsy was used. Considering that in the main text is clear the workflow, try to briefly summarize it also in the abstract.

26-27: We thank the Reviewer for this suggestion. Reading your advice, we modified the phrase “Our study aimed to identify specific serum metabolites correlated with Tacrolimus toxicity” with “Our study aimed to identify specific serum metabolites correlated with high tacrolinemia” so it would be understandable that we only used serum samples in order to avoid invasive methods as graft biopsy.

RESULTS

194: This is not reported in the workflow, which is, fairly, generic.

257-258: We thank the Reviewer for this observation. We deleted the part with the workflow correlation, so it remains as a statement about what we did:

“The student’s t-test and ROC analysis were used for the selection of the five statistical relevant metabolites (Table 2).”

228-231: Check sentences. Some were referred to a table probably.

291-301: We thank the Reviewer for this observation. Regarding those sentences, they are the legend of the table above just like the ones below the table before: Table 4 and Table 5.

REFERENCES

Try to use references not older than 2010.

We thank the Reviewer for this suggestion. About the few references that we have before 2010 we consider them relevant for our research, because to our knowledge there aren’t newer studies about the same issue.

Round 2

Reviewer 2 Report

No further corrections required.